# Association between Reproductive Span and Sarcopenia

**DOI:** 10.3390/ijerph18010154

**Published:** 2020-12-28

**Authors:** Eun Young Park, Kyoung Hee Han, Tae Ha Chung, Nam Yun Kim, Ji Min Lee, Seong Jin Choi, Jong Koo Kim

**Affiliations:** 1Department of Obstetrics and Gynecology, Wonju College of Medicine, Yonsei University, Wonju 26426, Korea; evenezer@yonsei.ac.kr (E.Y.P.); Visionhk@yonsei.ac.kr (K.H.H.); rlaskafus493@gmail.com (N.Y.K.); latteqd@gmail.com (J.M.L.); 2Department of Family Medicine, Wonju College of Medicine, Yonsei University, Wonju 26426, Korea; medeus115@yonsei.ac.kr

**Keywords:** reproductive span, menarche, menopause, sarcopenia

## Abstract

Sarcopenia is defined as an age-related loss of skeletal muscle and is associated with several health disorders. Causes of sarcopenia, which included physical inactivity, alcohol, dietary habits, and smoking, have been researched. The present study was undertaken to examine the association between reproductive span and sarcopenia in Korean women. Data obtained from 2008 to 2011 Korea National Health and Nutrition Examination Surveys (KNHANES) were analyzed. We defined sarcopenia based on the cut-off values of the Foundation for the National Institutes of Health (FNIH) sarcopenia project criteria: ASM/BMI < 0.512 for women. Reproductive span was defined as years from menarche to menopause, and we divided the 3970 study subjects into three groups by reproductive span tertile. Multivariate logistic regression analysis was used to determine adjusted ORs for the relation between reproductive span and sarcopenia. The prevalence of sarcopenia in the study was 17.7% (704 of 3970). Multiple logistic regression analysis was performed using weighted populations. After adjusting for covariates, reproductive span was found to be inversely associated with the risk of sarcopenia [Tertile 1 = 1 (reference); Tertile 2, odds ratio (OR) = 0.927, 95% confidence interval (CI) = 0.863–0.995; Tertile 3, OR = 0.854, 95% CI = 0.793–0.915].

## 1. Introduction

Sarcopenia was first described by Irwin Rosenberg in 1989, and is characterized by age-related loss of skeletal muscle mass, strength, and function [1,2]. Sarcopenia is associated with health problems, such as mobility disorders, increased fall risk, reduced ability to perform daily activities, and loss of independence [3]. Several contributing factors for sarcopenia have been reported. A study analyzing participants living in New Mexico reported that physical disability was a risk factor for sarcopenia [3]. Lau et al. [4] recruited young and healthy Chinese participants living in Hong Kong and reported that underweight (low BMI) was significantly related with an increased ratio of sarcopenia. Research by Landi et al. [5] analyzing participants in the nursing home of Rome reported that chronic disease, such as cardiovascular disease (CVD) and osteoarthritis (OA), low BMI, and lesser involvement in leisure physical activity were significantly associated with increased risk of sarcopenia.

Menarche and menopause are two indicators of reproductive history, and associations between them and sarcopenic status have been well investigated [6,7,8,9,10]. During puberty, elevated levels of sex steroids, growth hormones, and vitamin D levels and dietary and basal metabolic rate increases result in remarkable muscle growth [7,11]. With regard to menopause, the majority of studies conducted on the subject have concluded that menopause is independently associated with muscle loss after adjusting for age [9,10], and several authors have suggested that loss of estrogen is primarily responsible for this menopause-induced reduction in muscle mass [11,12].

Although many studies have addressed relations between reproductive indicators and muscle in women, no study has previously investigated the relation between reproductive span and sarcopenia. Accordingly, in the present study, we aimed to identify the association between reproductive span and sarcopenia in Korean women using data obtained during the 2008–2011 Korean National Health and Nutrition Examination Surveys (KNHANES).

## 2. Materials and Methods

### 2.1. Study Population

KNHANES surveys have been conducted annually using a cross-sectional, rolling sample and stratified cluster design by the Division of Chronic Disease Surveillance of Korea Center for Disease Control and Prevention since 1998. The following exclusion criteria were applied: subjects with incomplete answers about social and medical histories; incomplete answers about reproductive history (years of menarche, menopause); a history of cervical cancer or total hysterectomy, and lack of blood pressure (BP), weight, height, waist circumference (WC), or laboratory marker data. From all survey participants in the 2008–2011 KNHANES surveys, 3970 postmenopausal women aged above 40 years were recruited for this study. All participants provided written informed consent and data were recorded anonymously.

### 2.2. Variable Measurement

We used the International Physical Activity Questionnaire short form (IPAQ-SF) to estimate the overall physical activity (PA) amount of a participant by the duration (in hours) and number of days (in a week) of engagement in three types of activities (walking, moderate intensity activities, and vigorous intensity activities) in the past 7 days [13,14]. We measured the amount of PA based on metabolic equivalent (MET), which is a unit to estimate the amount of oxygen used by the body during exercise or physical activity. In detail, we calculated MET-h/week by multiplying the MET score of an activity (4.0 for moderate intensity PA and 8.0 for vigorous PA) and the hours and days of exercising. 

We measured the amount of smoking by unit, pack-years, and the amount of alcohol based on grams per week. Total energy (kcal/day) consumptions were calculated using a food composition table produced and validated by Rural Development Administration [15].

Height was measured (in centimeters) using a wall-mounted measuring scale, and weight was measured (in kilograms) using a calibrated electronic scale with subjects wearing light clothing without shoes. Body mass index (BMI) was calculated using weight (kg)/height^2^ (m^2^). BP was measured by well-trained observers using a mercury sphygmomanometer (Baumanometer; WA Baum Co., Inc., Copiague, NY, USA) after subjects had rested for 5 min in a seated position. Final BP values were calculated by averaging the second and third measurements. 

Hypertension was defined according to JNC-8 as systolic BP of >140 mmHg or diastolic BP of >90 mmHg or treatment with antihypertensive medication [16]. Diabetes was defined as fasting blood glucose of >126 mg/dL or treatment with glucose-lowering drugs. Blood samples were collected by venipuncture after fasting for more than 10 h. Serum levels of fasting glucose, total cholesterol (TC), triglyceride (TG), and creatinine (Cr) were measured using a Hitachi automatic analyzer 7600 (Hitachi, Tokyo). Samples were immediately refrigerated, transported to the Central Testing Institute (Seoul) and analyzed within 24 h.

### 2.3. Definition of Sarcopenia and Reproductive Span

Dual-energy X-ray absorptiometry (DXA, Lunar Corp., Madison, WI, USA) was used to measure body compositions. Several definitions of sarcopenia have been proposed. We used BMI-adjusted appendicular skeletal muscle mass (ASM), which was calculated by dividing ASM (defined as summed muscle mass of arms and legs) by BMI. We defined sarcopenia based on the cut-off values of the Foundation for the National Institutes of Health (FNIH) sarcopenia project criteria: ASM/BMI < 0.512 for women [17]. Furthermore, we used another definition for sarcopenia proposed by Asian Working Group for Sarcopenia (AWGS) 2019: ASM/height^2^ < 5.4 [18].

We excluded subjects with menarche age <10 years and menopause age >60 years to reduce effects of outliers. Reproductive span was calculated from age at menarche to age at menopause for each of the study subjects, which were divided into three groups based on reproductive span tertiles.

### 2.4. Statistical Analyses

Statistical analysis was performed using SPSS ver. 20.0 (IBM Corp., Armonk, NY, USA). To calculate the total population represented by the 3970 study subjects, we employed the stratification variables and sampling weights. The aim of the sampling weight usage was to ensure that calculated estimates or results are truly representative of the Korean population [19]. Sampling weights are estimated by three factors: probabilities of sample selection, adjustment for nonresponse, and a post-stratification factor [19]. We used the sampling weights that were previously calculated by the Korean Center for Disease Control and Prevention.

Data distributions were found to be normal, which allowed the use of parametric tests. Continuous variables such as age, BP, and laboratory markers, were analyzed by one-way ANOVA followed by Tukey’s multiple comparison. For categorical variables, the chi-square test was used to compare frequencies between groups followed by the multiple comparison of Benjamini–Hochberg. Binary logistic regression analysis was performed to examine odds ratios (ORs) of sarcopenia covariates. Multivariate logistic regression analysis was used to determine the adjusted ORs of reproductive span for sarcopenia. *p*-values of <0.05 were considered statistically significant.

## 3. Results

A total of 3970 women, who were estimated as a population of approximately 16 million people after weighting, were included in this study. The general characteristics of study subjects classified by reproduction span tertile are summarized in Table 1. Reproductive spans ranged from age 5 to 50 years. The prevalence of sarcopenia among study subjects was 17.7% (704 of 3970). The means of ASM, ASM/BMI, and ASM/height^2^ were greatest in the longest reproductive span group (tertile 3). Furthermore, the prevalence of sarcopenia was highest in the tertile 1 group. Subjects in tertile 3 had the following characteristics when compared to those in tertile 1 or 2: young age, high ratio of hypertension and dyslipidemia medication, high BMI, high diastolic BP, low white blood cell (WBC) count, high total energy and protein intake.

Binary and multiple logistic regression analyses were performed using the weighted population. The weighting values that were applied in this study were calculated by KNHANES. Univariate logistic regression showed that the risk of sarcopenia as defined by FNIH [17] increased with shorter reproductive span, misuse of oral contraceptive and hormone replacement therapy (HRT), older age, lower amounts of smoking and alcohol consumption, less PA, the presence of hypertension, diabetes, and dyslipidemia, higher levels of BMI, WC, systolic BP, WBC count, fasting glucose, TC, TG, and Cr, less total energy and protein intakes (Table 2). In case of the risk of sarcopenia defined by AWGS 2019 [18], the following factors had similar patterns with those by FINH: ages at menarche and menopause, reproductive span, age, alcohol consumption, PA, WBC count, TC, total energy and protein intake (Table 2).

We selected covariates significantly associated with the risk of sarcopenia by binary logistic regression analysis (Table 3). Age-adjusted ORs (Model 1 in Table 3) showed that a longer reproductive span was associated with a lower prevalence of sarcopenia as defined by FNIH [17]. After adjusting for all selected covariates, reproductive span was found to be inversely associated with the risk of sarcopenia by FNIH (Table 3). When comparing sarcopenia as defined by AWGS 2019 [18] with that by FNIH, convergent results were proposed (Table 3).

Figure 1 illustrates correlation between reproductive span and the amount of skeletal muscle mass. Longer reproductive span was significantly associated both with more ASM (Pearson correlation coefficient (PCC) = 0.0799, *p* < 0.001) and ASM/BMI (PCC = 0.0444, *p* < 0.001, Figure 1). After adjusting ASM by age, the associations of reproductive span with ASM (PCC = 0.216, *p* < 0.001) and ASM/BMI (PCC = 0.207, *p* < 0.001) were strengthened (Figure 1).

## 4. Discussion

The present study demonstrates that reproductive span is significantly associated with a reduced risk of developing sarcopenia in the Korean population. Although the prevalence of sarcopenia did not differ between tertile 2 and tertile 3 groups, after applying weighting values and adjusting for covariates, the prevalence of sarcopenia was found to be significantly associated with reproductive span. Shafiee et al. [20] reviewed 18 studies analyzing sarcopenia diagnosed by DXA, and reported that the prevalence of sarcopenia ranged from 4.1 to 36.6%. Furthermore, this study estimated the overall prevalence of sarcopenia diagnosed by DXA or bioelectrical impedance analysis as 10% (95% CI: 8–12%) in men and 10% (95% CI: 8–13%) in women [20]. Our study reported 17.7% as the prevalence of sarcopenia, which was somewhat high compared to previous studies [20].

In addition, the present study showed that older age at menopause reduced the risk of sarcopenia. In a cross-sectional French study conducted on 205 healthy white women aged 45 to 70 years, it was observed that those aged 45 to 56 years at menopause had lower mean leg muscle mass (11.1 kg vs. 11.7 kg) than those aged 46 to 70 at menopause [8]. Tom et al. [21] analyzed 1765 participants aged ≥ 60 in NHANES III study and reported that women that underwent menopause at an early age due to surgery showed significantly diminished physical functions as compared with the natural menopausal group, which suggested that women with an older age at menopause have better subsequent physical functioning due to a longer period of exposure to endogenous hormones.

Menopause is characterized by many hormonal changes, especially estrogen changes [21]. When menopause starts, estrogen levels fall to about 50% of premenopausal levels [21]. Furthermore, changes in endocrine function due to reductions in estrogen and growth hormone levels cause type II fiber and motor unit losses and intramuscular fat accumulation, which have been shown to be related to the risk of sarcopenia [10]. Notably, in a randomized, double-blind, crossover study conducted on 16 healthy postmenopausal women that underwent hormone replacement therapy or placebo treatment, lean body mass increased and decreased, respectively [22].

The average ages at menarche were 16.74, 16.31, 16.18, 15.47, 15.13, and 14.39 for Korean women born in 1940–1944, 1945–1949, 1950–1954, 1955–1959, 1960–1964, and 1965–1969, respectively [23]. According to the duration of 1940–1969, the mean age at menarche in Korean women was 15.7 [23]. The average ages at menarche were 13.8 for Japanese women born in the 1930s, 13.3 for the 1940s, 12.8 for the 1950s, 12.3 for the 1960s (*n* = 20,547), 12.2 for the 1970s (*n* = 6568), and 12.2 for the 1980s (*n* = 281), respectively, and the overall average age at menarche was 12.6 years [24]. The average ages at menarche were 14.39 and 13.18 for Indonesian women born in the 1944 and 1988, respectively [25]. Our subjects had similar trends with ages at menarche in Korean women born in 1940–1959 [23], and somewhat late ages at menarche compared to those in Japanese and Indonesian women (Table 1). 

In the present study, an older age at menarche was associated with the risk of sarcopenia (Table 2). In a cross-sectional study performed on 97 female students in Tanzania, body fat percentage was 22% greater in postmenarche girls than in premenarche girls and age at menarche and body muscle percentage were negatively correlated (beta-coefficient −0.12) [6]. A cross-sectional study by Gemelli et al. [26] reported that girls with experience of menarche had more mass of skeletal muscle than those without menarche.

With aging in menopausal women, it was reported that atrophy and denervation of type II fibers occurred [27,28]. Widrick et al. [29] reported convergent results that type II fibers occupied more area than type I fiber in postmenopausal women. From these results, we causally suggested muscle redistribution in postmenopausal women as the possible mechanism for sarcopenia. Furthermore, in menopausal women, lipoprotein lipase activity was reported to increase in both the abdominal and the gluteal regions, inducing fat accumulation in the form of intramuscular fat [30,31,32]. This might lead to a higher risk for metabolic syndrome in postmenopausal women with a short duration of reproductive span [33]. Recently, Shin and Kwon reported that a short reproductive span had a causal relationship with increased risk of insulin resistance [34].

Moreover, reproductive span could be considered to reflect cumulative exposure to sex hormones like estrogen, which are known to have systemic anti-inflammatory effects [12]. Direct evidence of the protective effect of estrogen was provided in an animal model, in which 17 β-estradiol administered parenterally to ovariectomized monkeys resulted in 50–70% decreases in coronary artery atherosclerosis [35]. The mechanism underlying the protective effects of estrogen on vessels has been reported to involve fibroblast growth factor-2, vascular endothelial growth factor, and nitric oxide synthesis [36,37,38], and recently, reproductive span was demonstrated to be inversely related to cardiovascular disease [39] and metabolic syndrome [33]. We cautiously suggest that these findings and our finding that reproductive span is significantly and inversely association with sarcopenia indicate that a lack of estrogen and systemic and vessel inflammation underlie these associations.

To the best our knowledge, this study is the first large-scale study analyzing the association between reproductive span and sarcopenia in Korea. Furthermore, the weighting exercise indicates that it well represents the Korean population, and we adjusted for several confounders, such as age, socioeconomic status, smoking, physical activity, and medical history. 

Nevertheless, the study has some limitations. First, because of its cross-sectional nature, the causal relationship between reproductive span and sarcopenia could not be accessed. Second, the 2008–2011 KNHANES only measured muscle mass using a DXA, but did not measure strength and performance. Future study is needed to analyze the association of reproductive indices with sarcopenia diagnosed by a broader definition. Third, the study is subject to recall bias because ages at menarche and menopause were collected using self-reported questionnaires, although previous studies have shown the validity and reproducibility of self-reported age at menopause and menarche are fairly good [40,41]. Fourth, serum levels of hormones such as estrogen, luteinizing hormone, and follicular stimulating hormone were not included, although it has been reported that self-reported ages at menarche and menopause are highly correlated with menopause status as confirmed by estrogen levels [42]. Fifth, our study included only the Korean population, and further research that involves a multiethnic population is needed to propose generalized results.

## 5. Conclusions

Collectively, we conclude that a longer reproductive span is significantly associated with a lower risk of sarcopenia. We believe that better understanding of the association between reproductive history and muscle mass is likely to lead to better management of sarcopenia and early menopause in women, and suggest prospective studies be undertaken to determine the effects of reproductive span on sarcopenia and to elucidate the physiologic mechanisms responsible. Furthermore, in the viewpoint of public health, our study proposed women’s reproductive indices such as ages at menarche and menopause and their integrative index (reproductive span) that are easily obtained from a questionnaire as the early screening markers for sarcopenia.

## Figures and Tables

**Figure 1 ijerph-18-00154-f001:**
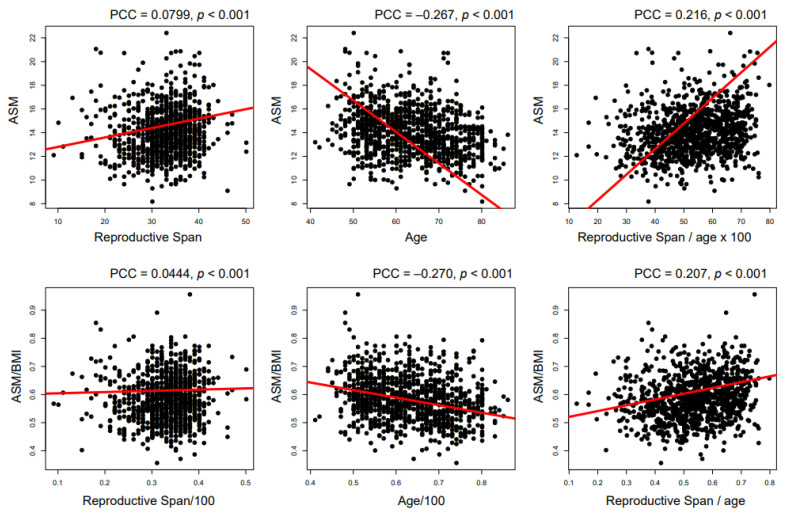
Association between reproductive span and muscle amount. (ASM, appendicular skeletal muscle; BMI, body mass index; PCC, Pearson correlation coefficient).

**Table 1 ijerph-18-00154-t001:** General characteristics of study subjects according to reproductive span.

	Reproductive Span, Years			*p*-Value
	≤31 (Tertile 1)	32–35 (Tertile 2)	≥36 (Tertile 3)	
Numbers	1373	1355	1242	
Menarche age, years	16.7 ± 0.05	15.9 ± 0.05	15 ± 0.05	<0.001 ^a,b,c^
Menopausal age, years	43.6 ± 0.12	49.5 ± 0.05	53.1 ± 0.07	<0.001 ^a,b,c^
Use of oral contraceptive, n	309 (22.5)	320 (23.6)	298 (23.9)	0.641
Use of HRT, n	214 (15.6)	218 (16.1)	210 (16.9)	0.653
ASM, kg	13.9 ± 0.05	14 ± 0.05	14.3 ± 0.05	<0.001 ^b,c^
ASM adjusted by BMI, 1/m^2^	0.5794 ± 0.0022	0.5869 ± 0.0021	0.5875 ± 0.0022	0.012 ^a,b^
ASM adjusted by height^2^, kg/m^2^	5.96 ± 0.019	5.93 ± 0.018	6.00 ± 0.019	0.038 ^c^
Sarcopenia by ASM/BMI, n	277 (20.2)	226 (16.7)	201 (16.1)	0.013 ^a,b^
Sarcopenia by ASM/Height^2^, n	292 (21.3)	282 (20.8)	223 (18)	0.076
Age, years	65 ± 0.27	62.3 ± 0.24	62.1 ± 0.22	<0.001 ^a,b^
Smoking, pack years	1.7 ± 0.2	1 ± 0.16	1.1 ± 0.17	0.015 ^a^
Alcohol consumption, g/week	13.3 ± 1.49	11.7 ± 1.21	9.7 ± 1.07	0.132
PA, MET-h/week	47.8 ± 1.98	46.5 ± 1.94	47.4 ± 2.1	0.891
Hypertension, n	719 (52.4)	670 (49.4)	673 (54.2)	0.05
Diabetes, n	230 (16.8)	192 (14.2)	178 (14.3)	0.111
Dyslipidemia medication, n	133 (9.7)	135 (10)	164 (13.2)	0.006 ^b,c^
BMI, kg/m^2^	24.1 ± 0.09	24.1 ± 0.09	24.5 ± 0.09	0.002 ^b,c^
Waist circumference, cm	82.3 ± 0.25	82.2 ± 0.25	82.8 ± 0.26	0.186
Systolic BP, mmHg	128.4 ± 0.49	126.6 ± 0.5	128 ± 0.5	0.023 ^a^
Diastolic BP, mmHg	77.2 ± 0.27	77.5 ± 0.27	78.8 ± 0.28	<0.001 ^b,c^
White blood cell count, 103/µL	6 ± 0.05	5.7 ± 0.04	5.6 ± 0.04	<0.001 ^a,b^
Hemoglobin	13.1 ± 0.03	13.1 ± 0.03	13.2 ± 0.03	0.125
Fasting glucose, mg/dL	101.5 ± 0.67	99.6 ± 0.59	101.2 ± 0.65	0.076
Total cholesterol, mg/dL	200 ± 1	202.2 ± 1	201.3 ± 1.05	0.281
Triglyceride, mg/dL	142.7 ± 2.62	137.4 ± 2.3	137.2 ± 2.42	0.2
Creatinine	0.7312 ± 0.0054	0.7282 ± 0.0049	0.724 ± 0.0048	0.602
Total energy (kcal/day)	1503.9 ± 14.9	1574.1 ± 16.14	1631.6 ± 18.47	<0.001 ^a,b,c^
Protein, g/week	49.1 ± 0.71	53.1 ± 0.73	55.6 ± 0.79	<0.001 ^a,b,c^

Results are presented as means ± standard errors or numbers (%) for categorical variables; ^a^
*p* < 0.05 Tertile 1 vs. 2 by post hoc comparison: [ANOVA: Tukey HSD; Chi-squared test: false discovery rate (FDR) adjustment]; ^b^
*p* < 0.05 Tertile 1 vs. 3 by post hoc comparison: (ANOVA: Tukey HSD; Chi-squared test: FDR adjustment); ^c^
*p* < 0.05 Tertile 2 vs. 3 by post hoc comparison: (ANOVA: Tukey HSD; Chi-squared test: FDR adjustment). HRT, hormone replacement therapy; ASM, appendicular skeletal muscle mass; BMI, body mass index; PA, physical activity; BP, blood pressure.

**Table 2 ijerph-18-00154-t002:** Binary logistic regression results for covariates of sarcopenia.

	OR (95% CI) for Sarcopenia by ASM/BMI	*p*-Value	OR (95% CI) for Sarcopenia by ASM/Height^2^	*p*-Value
Age at menarche	2.025 (1.742–2.354)	<0.001	1.537 (1.337–1.768)	<0.001
Age at menopause	0.604 (0.513–0.71)	<0.001	0.64 (0.55–0.746)	<0.001
Reproductive span, years	0.977 (0.972–0.981)	<0.001	0.98 (0.975–0.985)	<0.001
Use of oral contraceptive	0.879 (0.823–0.939)	<0.001	1.076 (1.014–1.142)	0.015
Use of HRT	0.579 (0.532–0.629)	<0.001	1.245 (1.166–1.329)	<0.001
Age, years	1.057 (1.054–1.06)	<0.001	1.007 (1.004–1.01)	<0.001
Smoking, pack years	0.926 (0.9–0.953)	<0.001	1.099 (1.075–1.123)	<0.001
Alcohol consumption, g/week	0.95 (0.937–0.963)	<0.001	0.976 (0.964–0.988)	<0.001
PA, MET-h/week	0.905 (0.895–0.916)	<0.001	0.938 (0.928–0.948)	<0.001
Hypertension	2.114 (1.998–2.236)	<0.001	0.781 (0.742–0.821)	<0.001
Diabetes	1.488 (1.388–1.596)	<0.001	0.946 (0.881–1.016)	0.13
Dyslipidemia medication	1.299 (1.196–1.41)	<0.001	0.524 (0.475–0.577)	<0.001
BMI, kg/m^2^	1.252 (1.241–1.263)	<0.001	0.626 (0.618–0.634)	<0.001
Waist circumference, cm	1.064 (1.061–1.067)	<0.001	0.897 (0.894–0.901)	<0.001
Systolic blood pressure, mmHg	4.163 (3.645–4.754)	<0.001	0.714 (0.632–0.807)	<0.001
Diastolic blood pressure, mmHg	1.065 (0.922–1.229)	0.392	0.429 (0.375–0.49)	<0.001
White blood cell count, 10^3^/µL	1.296 (1.275–1.317)	<0.001	1.072 (1.056–1.089)	<0.001
Hemoglobin	1.004 (0.977–1.031)	0.776	0.877 (0.855–0.898)	<0.001
Fasting glucose, mg/dL	1.777 (1.619–1.951)	<0.001	0.832 (0.754–0.918)	<0.001
Total cholesterol, mg/dL	1.718 (1.548–1.906)	<0.001	1.213 (1.102–1.336)	<0.001
Triglyceride, mg/dL	1.343 (1.297–1.39)	<0.001	0.852 (0.824–0.88)	<0.001
Creatinine	1.315 (1.153–1.499)	<0.001	0.951 (0.825–1.097)	0.49
Total energy (kcal/day)	0.602 (0.575–0.631)	<0.001	0.709 (0.679–0.74)	<0.001
Protein, g/week	0.658 (0.635–0.683)	<0.001	0.788 (0.762–0.815)	<0.001

Binary logistic regression was used for the weighted population; ASM, appendicular skeletal muscle mass; BMI, body mass index; OR, odds ratio; CI, confidence interval.

**Table 3 ijerph-18-00154-t003:** Multivariate logistic regression analysis of the relation between reproductive span and sarcopenia.

Reproductive Span, Years				
	Tertile 1 ≤31 Reference	Tertile 2 32–35 OR (95% CI)	Tertile 3 ≥36 OR (95% CI)	*p* for Trend
Sarcopenia by ASM/BMI				
Model 1	1	0.931 (0.872–0.995)	0.878 (0.820–0.941)	<0.001
Model 2	1	0.920 (0.860–0.984)	0.888 (0.828–0.953)	0.001
Model 3	1	0.927 (0.863–0.995)	0.854 (0.793–0.920)	<0.001
Sarcopenia by ASM/height^2^				
Model 1	1	0.933 (0.879–0.991)	0.736 (0.690–0.784)	<0.001
Model 2	1	0.949 (0.893–1.009)	0.789 (0.740–0.842)	<0.001
Model 3	1	0.950 (0.886–1.018)	0.889 (0.826–0.958)	0.002

Multiple logistic regression was used for the weighted population; Model 1: adjusted by age; Model 2: Model 1 + adjusted by hypertension, diabetes, dyslipidemia medication, regular exercise, alcohol consumption, current smoking, total energy, protein intake, use of oral contraceptive, and use of hormone replace therapy; Model 3: Model 2 + adjusted by waist circumference, systolic and diastolic blood pressure, total cholesterol, triglyceride, fasting glucose, creatinine, white blood cell, and hemoglobin; OR, odds ratio; CI, confidence interval.

## Data Availability

The data that support the findings of this study are available from the is publicly available at https://knhanes.cdc.go.kr/knhanes/eng/.

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
