# Peer review of "Association between Reproductive Span and Sarcopenia"

_ijerph, 2020, doi:10.3390/ijerph18010154_

Round 1

Reviewer 1 Report

This is an interesting study on a relatively large population size in Korea, the observations are largely confirmatory with other work. Did the authors account for hormone replacement therapies in their cohort. Further, can the authors comment on the applicability of or contrast with studies in other ethnicities?

Author Response

Answers to Reviewer #1

  1. This is an interesting study on a relatively large population size in Korea, the observations are largely confirmatory with other work. Did the authors account for hormone replacement therapies in their cohort. Further, can the authors comment on the applicability of or contrast with studies in other ethnicities?

Answer) Thank you for your comment. This comment included two tasks. First was about the HRT, which required all analyses in our study to be done from the beginning (data preprocessing), and we all agreed that it would be a worthwhile effort. Furthermore, five subjects did not have information about HRT, therefore, the number of all subjects to be analyzed changed from 3,975 to 3,970. After including HRT in our results, the association between reproductive span and sarcopenia was also significant. We described the changed in the Table 1, Table 2, and Table 3.

Second task was about the inclusion of other ethnicities and comparison of other results from other ethnicities. Unfortunately, the KNHANES only included the Korean population. Furthermore, there was no study reporting the relationship between reproductive span and sarcopenia. Therefore, we previously described the association of sarcopenia with age at menarche (Line 207 – 212) and age at menopause (Line 183 – 190). From these reasons, we described the second task as limitation contents.

For the First task

Line 126) Table 1

Table 1. General characteristics of the study subjects according to reproductive span.

Reproductive span, years

p-value

≤ 31

(Tertile 1)

32 – 35

(Tertile 2)

≥ 36

(Tertile 3)

Numbers

1373

1355

1242

Menarche age, years

16.7 ± 0.05

15.9 ± 0.05

15 ± 0.05

<0.001abc

Menopausal age, years

43.6 ± 0.12

49.5 ± 0.05

53.1 ± 0.07

<0.001abc

Use of oral contraceptive, n

309 (22.5)

320 (23.6)

298 (23.9)

0.641

Use of HRT, n

214 (15.6)

218 (16.1)

210 (16.9)

0.653

Line 144) Table 2

Table 2. Binary logistic regression results for covariates of sarcopenia.

OR (95 % CI)

for sarcopenia by ASM/BMI

p-value

OR (95 % CI)
for sarcopenia by ASM/height2

p-value

Age at menarche

2.025 (1.742 - 2.354)

<0.001

1.537 (1.337 - 1.768)

<0.001

Age at menopause

0.604 (0.513 - 0.71)

<0.001

0.64 (0.55 - 0.746)

<0.001

Reproductive span, years

0.977 (0.972 - 0.981)

<0.001

0.98 (0.975 - 0.985)

<0.001

Use of oral contraceptive

0.879 (0.823 - 0.939)

<0.001

1.076 (1.014 - 1.142)

0.015

Use of HRT, n

0.579 (0.532 - 0.629)

<0.001

1.245 (1.166 - 1.329)

<0.001

Line 154) Table 3

Table 3. Multivariate logistic regression analysis of the relation between reproductive span and sarcopenia.

Reproductive span, years

Tertile 1

≤ 31

Reference

Tertile 2

32 – 35

OR (95% CI)

Tertile 3

≥ 36

OR (95% CI)

p for trend

Sarcopenia by ASM/BMI

Model 1

1

0.931 (0.872 - 0.995)

0.878 (0.820 - 0.941)

<0.001

Model 2

1

0.920 (0.860 - 0.984)

0.888 (0.828 - 0.953)

0.001

Model 3

1

0.927 (0.863 - 0.995)

0.854 (0.793 - 0.920)

<0.001

Sarcopenia by ASM/height2

Model 1

1

0.933 (0.879 - 0.991)

0.736 (0.690 - 0.784)

<0.001

Model 2

1

0.949 (0.893 - 1.009)

0.789 (0.740 - 0.842)

<0.001

Model 3

1

0.950 (0.886 - 1.018)

0.889 (0.826 - 0.958)

0.002

Multiple logistic regression was used for the weighted population
Model 1: adjusted by age
Model 2: Model 1 + adjusted by hypertension, diabetes, dyslipidemia medication, regular exercise, alcohol consumption, current smoking, total energy, protein intake, use of oral contraceptive, and use of hormone replace therapy
Model 3: Model 2 + adjusted by waist circumference, systolic and diastolic blood pressure, total cholesterol, triglyceride, fasting glucose, creatinine, white blood cell, and hemoglobin
OR, odds ratio; CI, confidence interval.

For the Second task

Line 246) Discussion section

Fifth, our study included only Korean population, and further research that involved multi-ethnic population were needed to propose the generalized results.

Reviewer 2 Report

The authors present an observational study, which aims to assess the association between reproductive span and sarcopenia, using data from the KNHANES study, and using weighted populations. Their finding that a longer reproductive span is associated with lower risk of sarcopenia is an important and relevant one.

Specific points:

On line 84 it is mentioned that several definitions of sarcopenia have been proposed, why did the authors choose FNIH? More recent sarcopenia guidance suggests that muscle strength is a more important measure of sarcopenia than muscle mass (Cruz-Jentoft, 2019). Why was mass chosen? Was any data on muscle strength available? If not – this would be an important point for the discussion. However, if strength data is available it would be really interesting to see the analysis which includes it.

The prevalence of sarcopenia is 17.7% which is quite high – please include some discussion around this.

Ln 35: the authors present causes of sarcopenia. The aetiology of sarcopenia is generally thought of as a complex multi-factorial process, to which these named ‘causes’ have been reported as likely contributing factors. I would reconsider rephrasing this section.

Section 2.4 – please expand on methods for use of weighted populations.

Ln 129 – I believe there is some typo with which table the text is referring to

Ln 153: typo

Ln 159: sarah et al – does not correspond with names in ref 11

Ln 161 – physical functions is a bit vague – could you be more specific here?

The discussion explores oestrogen and briefly touches on inflammation as possible causes for the associated between reproductive span and sarcopenia. Are there any other possibilities or are these the only mechanisms with any suggestive evidence?

Author Response

Answers to Reviewer #2

  1. On line 84 it is mentioned that several definitions of sarcopenia have been proposed, why did the authors choose FNIH? More recent sarcopenia guidance suggests that muscle strength is a more important measure of sarcopenia than muscle mass (Cruz-Jentoft, 2019). Why was mass chosen? Was any data on muscle strength available? If not – this would be an important point for the discussion. However, if strength data is available it would be really interesting to see the analysis which includes it.

Answer) Thank you for your impressive comment. We used the 2008 – 11 KNHANES. Unfortunately, these datasets did not include information about muscle strength or quality. However, it was possible to include AWGS 2019 criteria. After applying the AWGS criteria, we also resulted consistent associational result between reproductive span and sarcopenia with those from the FNIH. We described the changed in the method, Table 1, Table 2, and Table 3.

Line 96) Methods section

Furthermore, we used another definition for sarcopenia proposed by Asian Working Group for Sarcopenia (AWGS) 2019: ASM/height2 < 5.4 [18].

Line 140) Result section

In case of the risk of sarcopenia defined by AWGS 2019 [18], following factors had similar patterns with those by FINH: ages at menarche and menopause; reproductive span; age; alcohol consumption; PA; WBC count; TC; total energy and protein intakes (Table 2).

Line 152) Result section

Comparing sarcopenia defined by AWGS 2019 [18] with those by FNIH, convergent results were proposed (Table 3).

Line 126) Table 1

Table 1. General characteristics of the study subjects according to reproductive span.

Reproductive span, years

p-value

≤ 31

(Tertile 1)

32 – 35

(Tertile 2)

≥ 36

(Tertile 3)

Numbers

1373

1355

1242

Menarche age, years

16.7 ± 0.05

15.9 ± 0.05

15 ± 0.05

<0.001abc

Menopausal age, years

43.6 ± 0.12

49.5 ± 0.05

53.1 ± 0.07

<0.001abc

Use of oral contraceptive, n

309 (22.5)

320 (23.6)

298 (23.9)

0.641

Use of HRT, n

214 (15.6)

218 (16.1)

210 (16.9)

0.653

ASM, kg

13.9 ± 0.05

14 ± 0.05

14.3 ± 0.05

<0.001bc

ASM adjusted by BMI, 1/m2

0.5794 ± 0.0022

0.5869 ± 0.0021

0.5875 ± 0.0022

0.012ab

ASM adjusted by height2, kg/m2

5.96 ± 0.019

5.93 ± 0.018

6.00 ± 0.019

0.038c

Sarcopenia by ASM/BMI, n

277 (20.2)

226 (16.7)

201 (16.1)

0.013ab

Sarcopenia by ASM/Height2, n

292 (21.3)

282 (20.8)

223 (18)

0.076

Age, years

65 ± 0.27

62.3 ± 0.24

62.1 ± 0.22

<0.001ab

Line 144) Table 2

Table 2. Binary logistic regression results for covariates of sarcopenia.

OR (95 % CI)

for sarcopenia by ASM/BMI

p-value

OR (95 % CI)
for sarcopenia by ASM/height2

p-value

Age at menarche

2.025 (1.742 - 2.354)

<0.001

1.537 (1.337 - 1.768)

<0.001

Age at menopause

0.604 (0.513 - 0.71)

<0.001

0.64 (0.55 - 0.746)

<0.001

Reproductive span, years

0.977 (0.972 - 0.981)

<0.001

0.98 (0.975 - 0.985)

<0.001

Use of oral contraceptive

0.879 (0.823 - 0.939)

<0.001

1.076 (1.014 - 1.142)

0.015

Use of HRT

0.579 (0.532 - 0.629)

<0.001

1.245 (1.166 - 1.329)

<0.001

Line 154) Table 3

Table 3. Multivariate logistic regression analysis of the relation between reproductive span and sarcopenia.

Reproductive span, years

Tertile 1

≤ 31

Reference

Tertile 2

32 – 35

OR (95% CI)

Tertile 3

≥ 36

OR (95% CI)

p for trend

Sarcopenia by ASM/BMI

Model 1

1

0.931 (0.872 - 0.995)

0.878 (0.820 - 0.941)

<0.001

Model 2

1

0.920 (0.860 - 0.984)

0.888 (0.828 - 0.953)

0.001

Model 3

1

0.927 (0.863 - 0.995)

0.854 (0.793 - 0.920)

<0.001

Sarcopenia by ASM/height2

Model 1

1

0.933 (0.879 - 0.991)

0.736 (0.690 - 0.784)

<0.001

Model 2

1

0.949 (0.893 - 1.009)

0.789 (0.740 - 0.842)

<0.001

Model 3

1

0.950 (0.886 - 1.018)

0.889 (0.826 - 0.958)

0.002

Multiple logistic regression was used for the weighted population
Model 1: adjusted by age
Model 2: Model 1 + adjusted by hypertension, diabetes, dyslipidemia medication, regular exercise, alcohol consumption, current smoking, total energy, protein intake, use of oral contraceptive, and use of hormone replace therapy
Model 3: Model 2 + adjusted by waist circumference, systolic and diastolic blood pressure, total cholesterol, triglyceride, fasting glucose, creatinine, white blood cell, and hemoglobin
OR, odds ratio; CI, confidence interval.

  1. The prevalence of sarcopenia is 17.7% which is quite high – please include some discussion around this.

Answer) We reviewed a meta-analysis paper analyzing about the prevalence of sarcopenia. This study reported the average estimated prevalence of sarcopenia was 10% in both men and women. Our study reported 17.7% of the prevalence of sarcopenia which was somewhat high value. We added above contents about comment and answer for the prevalence of sarcopenia in the manuscript as follows.

Line 178) Discussion section

Shafiee et al.[20] reviewed 18 studies analyzing sarcopenia diagnosed by DXA, and reported that the prevalence of sarcopenia ranged from 4.1 to 36.6%. Furthermore, this study estimated the overall prevalence of sarcopenia diagnosed by DXA or bio-electrical impedance analysis as 10% (95% CI: 8 – 12%) in men and 10% (95% CI: 8 – 13%) in women [20]. Our study reported 17.7% as the prevalence of sarcopenia, which was somewhat high values compared to previous studies [20].

  1. Ln 35: the authors present causes of sarcopenia. The aetiology of sarcopenia is generally thought of as a complex multi-factorial process, to which these named ‘causes’ have been reported as likely contributing factors. I would reconsider rephrasing this section.

Answer) As you commented, we rephrased this section as follows.

Line 35) Introduction section

Several contributing factors for sarcopenia have been reported. A study analyzing participants living in New Mexico reported that physical disability was a risk factor for sarcopenia [3]. Lau et al.[4] recruited young and healthy Chinese participants living in Hong Kong and reported that underweight (low BMI) was significantly related with the increased ratio of sarcopenia. A research by Landi et al.[5] analyzing participants in the nursing home of Rome reported that chronic disease, such as cardiovascular disease (CVD) and osteoarthritis (OA), low BMI, and the less involvement of leisure physical activity were significantly associated with the increased risk of sarcopenia.

  1. Section 2.4 – please expand on methods for use of weighted populations.

Answer) We described the detail contents about the weight values.

Line 102) Method section (Statistical analyses)

To calculate the total population represented by the 3,970 study subjects, we employed the stratification variables and sampling weights. The aim of the sampling weights usage is to ensure that calculated estimates or results are truly representative of the Korean population [19]. The sampling weights are estimated by three factors: the probabilities of sample selection, an adjustment for non-response, and a post-stratification factor [19]. We used the sampling weights that were previously calculated by the Korean Center for Disease Control and Prevention.

  1. Ln 129,– I believe there is some typo with which table the text is referring to

Ln 153: typo

Ln 159: sarah et al – does not correspond with names in ref 11

Answer) Thank you for the detail comment. We corrected all errors.

  1. Ln 161 – physical functions is a bit vague – could you be more specific here?

Answer) We measured the amount of physical activity as follows.

MET-hr/week =

We added the detail contents about the physical activity in the method section.

Line 67) Method section

We used the International Physical Activity Questionnaire-short form (IPAQ-SF) to estimate the overall physical activity (PA) amount of a participant by the duration (in hours) and number of days (in a week) of engagement in three types of activity (walking, moderate-intensity activities, and vigorous-intensity activities) in the past 7 days [13, 14].

  1. The discussion explores oestrogen and briefly touches on inflammation as possible causes for the associated between reproductive span and sarcopenia. Are there any other possibilities or are these the only mechanisms with any suggestive evidence?

Answer)

We considered three points, including muscle re-distribution, LPL activity, and insulin resistance as the possible mechanisms of reproductive indices for sarcopenia. With aging, an atrophy and denervation of type II fibers occur. As consistent results, Widrick et al. reported that the type II fibers occupied more area than type I fiber in the postmenopausal women. From these results, we suggested the muscle re-distribution in the postmenopausal women as the possible mechanism for sarcopenia. Furthermore, in the menopausal women, LPL activity was reported to increase in both abdominal and gluteal region, inducing an accumulation of fat as the form of intra-muscular fat. This might lead to a higher risk for diabetes mellitus in the post-menopausal women. Recently, Shin et al. reported that a short reproductive span had causal relationship with the increased risk of insulin resistance. We described above contents in the discussion section.

Line 213) Discussion section

With aging in menopausal women, it was reported that atrophy and denervation of type II fibers occurred [28, 29]. Widrick et al.[30] reported convergent results that the type II fibers occupied more area than type I fiber in the postmenopausal women. From these results, we causally suggested the muscle re-distribution in the postmenopausal women as the possible mechanism for sarcopenia. Furthermore, in the menopausal women, LPL activity was reported to increase in both abdominal and gluteal region, inducing the fat accumulation in the form of intra-muscular fat [31-33]. This might lead to a higher risk for metabolic syndrome in the post-menopausal women with short duration of reproductive span [34]. Recently, Shin and Kwon reported that a short reproductive span had causal relationship with the increased risk of insulin resistance [35].

Reviewer 3 Report

Comments to authors

This is an interesting article investigating the association between reproductive span and sarcopenia in Korean women. This is a well-written manuscript. Although several studies have published about relations between reproductive indicators and muscle in women, the study examined the relation between reproductive span and sarcopenia is lacking. Author also discussed the necessity of the proper management of sarcopenia and early menopause in women. However, there are some comments that I would like author to address.

Comment 1 :

Section of definition of sarcopenia in line 82.

As generally known about clinical diagnosis of sarcopenia, muscle quality should be measured in addition to muscle mass measurement. Most of current diagnostic tools include handgrip strength for measuring muscle strength and some physical functional examination.

As this study is cross-sectional, it could’ve been possible.

The prevalence of sarcopenia varies widely depending on the definition applied. Moreover, ethnic diversity in body composition should also be taken into consideration. To address these issues, the Asian Working Group for Sarcopenia (AWGS) proposed its own diagnostic criteria for people in Asia in 2014 and revised it in 2019. In this regard, did author have any idea of incorporating AWGS 2019 criteria for diagnosis of sarcopenia?

Comment 2 :

Table 1 in page 3, means of menarche age in each groups are 16.7 ± 0.05, 15.9 ± 0.05, and 15 ± 0.05. According to author, participants’ birth year are supposed to be between 1940’s and 1970’s as participants are postmenopausal women aged >40 years old. In this regard, mean menarche age in this study is unexpectedly higher than the reports published from other Asian countries such as Japan and Indonesia for instance. It is supposed to be around 12-13 year-old. I am convinced that there should be regional peculiarity. So, I suggest that it is better to mention about the trend of menarche age in Korea, especially the comparison to other countries.

Author Response

Answers to Reviewer #3

  1. Comment 1:

Section of definition of sarcopenia in line 82.

As generally known about clinical diagnosis of sarcopenia, muscle quality should be measured in addition to muscle mass measurement. Most of current diagnostic tools include handgrip strength for measuring muscle strength and some physical functional examination.

As this study is cross-sectional, it could’ve been possible.

The prevalence of sarcopenia varies widely depending on the definition applied. Moreover, ethnic diversity in body composition should also be taken into consideration. To address these issues, the Asian Working Group for Sarcopenia (AWGS) proposed its own diagnostic criteria for people in Asia in 2014 and revised it in 2019. In this regard, did author have any idea of incorporating AWGS 2019 criteria for diagnosis of sarcopenia?

Answer) Thanks for your impressive comment. Incorporating AWGS 2019 criteria to our study required all analyses in our study to be done from the beginning (data preprocessing), and we all agreed that it would be a worthwhile effort. After applying the AWGS criteria, we also resulted consistent associational result between reproductive span and sarcopenia with those from the FNIH. We described the changed in the method, Table 1, Table 2, and Table 3.

Line 96) Methods section

Furthermore, we used another definition for sarcopenia proposed by Asian Working Group for Sarcopenia (AWGS) 2019: ASM/height2 < 5.4 [18].

Line 140) Result section

In case of the risk of sarcopenia defined by AWGS 2019 [18], following factors had similar patterns with those by FINH: ages at menarche and menopause; reproductive span; age; alcohol consumption; PA; WBC count; TC; total energy and protein intakes (Table 2).

Line 152) Result section

Comparing sarcopenia defined by AWGS 2019 [18] with those by FNIH, convergent results were proposed (Table 3).

Line 126) Table 1

Table 1. General characteristics of the study subjects according to reproductive span.

Reproductive span, years

p-value

≤ 31

(Tertile 1)

32 – 35

(Tertile 2)

≥ 36

(Tertile 3)

Numbers

1373

1355

1242

Menarche age, years

16.7 ± 0.05

15.9 ± 0.05

15 ± 0.05

<0.001abc

Menopausal age, years

43.6 ± 0.12

49.5 ± 0.05

53.1 ± 0.07

<0.001abc

Use of oral contraceptive, n

309 (22.5)

320 (23.6)

298 (23.9)

0.641

Use of HRT, n

214 (15.6)

218 (16.1)

210 (16.9)

0.653

ASM, kg

13.9 ± 0.05

14 ± 0.05

14.3 ± 0.05

<0.001bc

ASM adjusted by BMI, 1/m2

0.5794 ± 0.0022

0.5869 ± 0.0021

0.5875 ± 0.0022

0.012ab

ASM adjusted by height2, kg/m2

5.96 ± 0.019

5.93 ± 0.018

6.00 ± 0.019

0.038c

Sarcopenia by ASM/BMI, n

277 (20.2)

226 (16.7)

201 (16.1)

0.013ab

Sarcopenia by ASM/Height2, n

292 (21.3)

282 (20.8)

223 (18)

0.076

Age, years

65 ± 0.27

62.3 ± 0.24

62.1 ± 0.22

<0.001ab

Line 144) Table 2

Table 2. Binary logistic regression results for covariates of sarcopenia.

OR (95 % CI)

for sarcopenia by ASM/BMI

p-value

OR (95 % CI)
for sarcopenia by ASM/height2

p-value

Age at menarche

2.025 (1.742 - 2.354)

<0.001

1.537 (1.337 - 1.768)

<0.001

Age at menopause

0.604 (0.513 - 0.71)

<0.001

0.64 (0.55 - 0.746)

<0.001

Reproductive span, years

0.977 (0.972 - 0.981)

<0.001

0.98 (0.975 - 0.985)

<0.001

Use of oral contraceptive

0.879 (0.823 - 0.939)

<0.001

1.076 (1.014 - 1.142)

0.015

Use of HRT

0.579 (0.532 - 0.629)

<0.001

1.245 (1.166 - 1.329)

<0.001

Line 154) Table 3

Table 3. Multivariate logistic regression analysis of the relation between reproductive span and sarcopenia.

Reproductive span, years

Tertile 1

≤ 31

Reference

Tertile 2

32 – 35

OR (95% CI)

Tertile 3

≥ 36

OR (95% CI)

p for trend

Sarcopenia by ASM/BMI

Model 1

1

0.931 (0.872 - 0.995)

0.878 (0.820 - 0.941)

<0.001

Model 2

1

0.920 (0.860 - 0.984)

0.888 (0.828 - 0.953)

0.001

Model 3

1

0.927 (0.863 - 0.995)

0.854 (0.793 - 0.920)

<0.001

Sarcopenia by ASM/height2

Model 1

1

0.933 (0.879 - 0.991)

0.736 (0.690 - 0.784)

<0.001

Model 2

1

0.949 (0.893 - 1.009)

0.789 (0.740 - 0.842)

<0.001

Model 3

1

0.950 (0.886 - 1.018)

0.889 (0.826 - 0.958)

0.002

Multiple logistic regression was used for the weighted population
Model 1: adjusted by age
Model 2: Model 1 + adjusted by hypertension, diabetes, dyslipidemia medication, regular exercise, alcohol consumption, current smoking, total energy, protein intake, use of oral contraceptive, and use of hormone replace therapy
Model 3: Model 2 + adjusted by waist circumference, systolic and diastolic blood pressure, total cholesterol, triglyceride, fasting glucose, creatinine, white blood cell, and hemoglobin
OR, odds ratio; CI, confidence interval.

Comment 2:

Table 1 in page 3, means of menarche age in each groups are 16.7 ± 0.05, 15.9 ± 0.05, and 15 ± 0.05. According to author, participants’ birth year are supposed to be between 1940’s and 1970’s as participants are postmenopausal women aged >40 years old. In this regard, mean menarche age in this study is unexpectedly higher than the reports published from other Asian countries such as Japan and Indonesia for instance. It is supposed to be around 12-13 year-old. I am convinced that there should be regional peculiarity. So, I suggest that it is better to mention about the trend of menarche age in Korea, especially the comparison to other countries.

Answer) As you commented, we reviewed two studies, one was about age at menarche in Japanese women (Secular trends in age at menarche and time to establish regular menstrual cycling in Japanese women born between 1930 and 1985) and the other was about that in Indonesian women (The trend in age at menarche in Indonesia: birth cohorts 1944–1988). Our subjects had similar trends with ages at menarche in Korean women born in 1940 – 59, and somewhat late ages at menarche compared to those in Japanese and Indonesian women. We described above contents in the discussion section.

Line 198) Discussion section

The average ages at menarche were 16.74, 16.31, 16.18, 15.47, 15.13, and 14.39 for Korean women born in the 1940 – 44, 1945 – 49, 1950 – 54, 1955 – 59, 1960 – 64, and 1965 – 1969, respectively [24]. According to the duration of 1940 – 1969, the mean age at menarche in Korean women was 15.7 [24]. The average ages at menarche were 13.8 for Japanese women born in the 1930s, 13.3 for the 1940s, 12.8 for the 1950s, 12.3 for the 1960s (n = 20,547), 12.2 for the 1970s (n = 6,568), and 12.2 for the 1980s (n = 281), respectively, and the overall average age at menarche was 12.6 years [25]. The average ages at menarche were 14.39 and 13.18 for Indonesian women born in the 1944 and 1988, respectively [26]. Our subjects had similar trends with ages at menarche in Korean women born in 1940 – 59 [24], and somewhat late ages at menarche compared to those in Japanese and Indonesian women (Table 1).
